# Boldine Alters Serum Lipidomic Signatures after Acute Spinal Cord Transection in Male Mice

**DOI:** 10.3390/ijerph20166591

**Published:** 2023-08-17

**Authors:** Zachary A. Graham, Jacob A. Siedlik, Carlos A. Toro, Lauren Harlow, Christopher P. Cardozo

**Affiliations:** 1Research Service, Birmingham VA Health Care System, Birmingham, AL 35233, USA; 2Healthspan, Resilience & Performance, Florida Institute for Human and Machine Cognition, Pensacola, FL 32502, USA; 3Department of Cell, Developmental, and Integrative Biology, University of Alabama-Birmingham, Birmingham, AL 35294, USA; 4Department of Exercise Science and Pre-Health Professions, Creighton University, Omaha, NE 68178, USA; jakesiedlik@creighton.edu; 5School of Medicine, Creighton University, Omaha, NE 68178, USA; 6Spinal Cord Damage Research Center, Bronx, NY 10468, USA; carlos.toro@mssm.edu (C.A.T.); lauren.harlow@va.gov (L.H.); christopher.cardozo@mssm.edu (C.P.C.); 7Icahn School of Medicine at Mount Sinai, New York, NY 10029, USA; 8Medical Service, James J. Peters VA Medical Center, Bronx, NY 10468, USA

**Keywords:** spinal cord injury, lipidomics, boldine

## Abstract

Traumatic spinal cord injury (SCI) results in wide-ranging cellular and systemic dysfunction in the acute and chronic time frames after the injury. Chronic SCI has well-described secondary medical consequences while acute SCI has unique metabolic challenges as a result of physical trauma, in-patient recovery and other post-operative outcomes. Here, we used high resolution mass spectrometry approaches to describe the circulating lipidomic and metabolomic signatures using blood serum from mice 7 d after a complete SCI. Additionally, we probed whether the aporphine alkaloid, boldine, was able to prevent SCI-induced changes observed using these ‘omics platforms’. We found that SCI resulted in large-scale changes to the circulating lipidome but minimal changes in the metabolome, with boldine able to reverse or attenuate SCI-induced changes in the abundance of 50 lipids. Multiomic integration using xMWAS demonstrated unique network structures and community memberships across the groups.

## 1. Introduction

Traumatic spinal cord injury (SCI) is a devasting event that leads to loss of sensation and voluntary movement, impaired bowel and bladder function, and the inability to properly regulate blood pressure and body temperature [1]. The first few days to weeks after SCI are also notable for unique metabolic disturbances that include transient hyperglycemia and other markers of metabolic dysregulation [2,3]. Our recent metabolomics analyses of mouse gastrocnemius muscle paralyzed by spinal cord transection revealed a transient decrease in tissue glucose levels at 7 d consistent with altered intramuscular regulation of glucose uptake and/or metabolism [4,5]. The relationship of the muscle metabolic profile after SCI and systemic metabolic function is likely related.

Improvements in mass spectrometry technology and methodological techniques, coupled with advances in bioinformatics, have greatly improved the resolution of detecting small compounds such as metabolites, including lipids, from biological tissues. These metabolomics-based approaches allow for identification of molecular signatures associated with a change to the physiological system that may identify a mechanism to target with an intervention. For example, a growing body of evidence indicates that circulating metabolites and lipids play important roles in organismal physiology and in responses to stresses such as exercise [6,7]. Understanding the effects of acute SCI on serum profiles of such signaling molecules can be accomplished in an unbiased manner using large-scale molecular phenotyping platforms based upon mass spectroscopy. Circulating factors from blood plasma/serum have been demonstrated to be affected by SCI severity in both pre-clinical and clinical models. In rats with mild or severe SCI, modeling of plasma metabolomic data from nuclear magnetic resonance (NMR) fingerprinting identified sets of metabolites associated with neural injury and energy dysregulation in the severe SCI group [8]. In humans with acute SCI (1–3 d post-injury), metabolite levels in the cerebrospinal fluid and serum were associated with severity of injury, with each tissue having potentially unique biomarker signatures [9]. Thus, high-resolution molecular phenotyping provides direction for identifying novel mechanisms behind cellular and organismal dysfunction to aid implementation of translatable therapies.

Finding efficacious molecular interventions that improve health and function after SCI remains a challenge. We recently demonstrated daily treatment of boldine, an aporphine alkaloid derived from the Chilean boldo tree (Peumus boldus), improves locomotor recovery, spares white matter, promotes axonal sprouting at the lesion site, and promotes large-scale changes in the spinal cord transcriptome associated with neurogenic recovery when compared to vehicle-treated mice across 28 d post-contusion SCI [10]. Boldine has anti-oxidative and anti-inflammatory actions that have been attributed to its ability to block connexin (Cx) hemichannels (HC) [11], which are non-selective pore proteins mainly localized to the cytoplasmic membrane that allow small molecules such as calcium ions, ATP, and glutamate to pass into and out of the cell. CxHC can form gap junctions (GJ) by binding to a CxHC on an adjacent cell thereby coupling cells electrically and chemically. Boldine does not block the formation or function of GJ [12], making its translatable potential more promising compared to other CxHC inhibitors.

Boldine may also affect skeletal muscle after SCI. CxHC are not normally present on the surface of skeletal muscle but appear de novo during denervation or various models of stress [13,14,15]. A critical role for CxHC in denervation atrophy was demonstrated by findings that genetic ablation of Cx43 and Cx45 in skeletal muscle blocked denervation atrophy [13] and that denervation-related changes in resting membrane potential and membrane permeability were blocked by CxHC blockers such as D4 [16]. The possibility that de novo sarcolemmal expression of CxHC contributes to muscle loss, dysfunction or metabolic derangements following SCI is suggested by the observation of increased membrane expression of Cx39, Cx43 and Cx45 in gastrocnemius muscle harvested at 56 d after complete spinal cord transection [13].

Further support for a role of de novo expression of CxHC in changes in skeletal muscle following SCI comes from our recent work that interrogated the transcriptomic, metabolomic, and DNA methylomic profiles of muscle paralyzed by SCI [5]. At the metabolomics level, boldine prevented or attenuated SCI-induced changes in the abundance of amino acids such as proline, phenylalanine, leucine, and isoleucine. Intriguingly, boldine seemed to block the fall in muscle levels of glucose, suggesting systemic metabolic function may be improved. This is supported by recent evidence that demonstrated boldine was able to improve systemic blood glucose levels in a streptozotocin-induced diabetes model in mice and rats [17]. We also noted boldine prevented an increase in muscle glutathione levels, suggesting a role in preventing generation of reactive oxygen species. Whether these effects of SCI on the skeletal muscle metabolome are related to changes in the serum metabolome/lipidome are unknown. Additionally, how boldine alters serum lipidomic and metabolomic profiles after SCI is also unknown. We therefore carried out untargeted metabolomics and lipidomics analysis of the serum of mice treated with boldine or vehicle that had been collected 7 d after a complete spinal cord transection. We then performed multiomic integration in an attempt to find multiomic interrelationships among groups. Our studies revealed large-scale changes to the circulating lipidome with slight changes in the metabolome. Importantly, boldine overturned and/or reduced changes in lipid levels after SCI. Additional multiomic integration studies demonstrated unique network structures.

## 2. Methods

### 2.1. Animals

We have previously published animal descriptions and treatment outcomes (e.g., weight, muscle loss, etc.) for this cohort of mice [5]. Briefly, 4-month-old C57BL/6 male mice were purchased from Charles River and kept in an AAALAC-accredited animal facility at the James J. Peters VAMC. C57BL/6 mice were used as they tolerate SCI well, are ubiquitously used across the scientific field, and make up the background for a large number of transgenic mice. Male mice were selected for the studies since males make up for ~80% of the total SCI population [18] and ~95% of the US Veteran population [19]. The 7 d timepoint was selected as we previously demonstrated this timepoint had greater responses in boldine-associated changes in the muscle metabolome and transcriptome following SCI [5]. Animals were kept on a standard light-dark cycle with ad libitum access to chow and water and were randomized to a laminectomy control group (Sham, n = 4) or laminectomy + T10 spinal cord transection with vehicle (SCIv, n = 6) or boldine (SCIb, n = 6) treatment. The study was reviewed and approved by the IACUC of the James J. Peters VAMC (Protocol CAR-16-54).

### 2.2. Laminectomy and Spinal Cord Transection

We have previously published detailed methods for mouse laminectomy and complete spinal cord transection surgeries [20]. In brief, mice were weighed then anesthetized using inhalation of 2–3% continuous-flow isoflurane. Hair along the back was shaved and the skin cleaned with 70% ethanol and betadine. An incision was made from T7-11 and the spinal column was exposed by blunt dissection and removal of the para-vertebral muscles. The vertebral arch of the T10 vertebral body was removed and the dura exposed. The incision site for the sham animals was then closed in layers using sutures for the muscle layer and surgical staples for skin. For the SCI animals, a micro-scalpel was passed through the spinal cord. Any residual tissue bridges were cut by a second pass of the scalpel when necessary. An inert gel foam was used to separate the severed spinal cord sections and the incision site was closed as described above. All animals were placed in a clean cage with Alpha dri+ bedding. Standard chow (Research Diets, D12450J: 10% fat, 20% protein, 70% carbohydrate; 3.82 kcal/g energy density) and Bio-Serv fruit treats were placed on the cage floor for easy access for all animals. All mice were single-housed for the remainder of the study.

### 2.3. Post-Operative Care and Boldine Administration

Animals were placed on 37 °C recirculating water heating pads for 24 h post-surgery. They received a cocktail of ketophen (5 mg/kg) and Baytril (5 mg/kg) subcutaneously daily for 3 d post-surgery, with a total daily volume of 1 mL of lactated Ringer’s, also administered subcutaneously, to prevent dehydration for all 7 d. Bladders were expressed 2 times per day. Boldine (50 mg/kg/d) was administered as previously reported [5,10] starting at 3 d post-injury. Briefly, boldine was dissolved in a mix of DMSO and peanut oil. The mixture was then added to peanut butter (PB) so that 1.0 g of total bolus of PB/boldine mix had the required dose of boldine. The final concentration of DMSO was less than 2.5%. Animals were familiarized with 1.0 g of PB for a week prior to surgery. All animals consumed 100% of their daily PB/boldine mix within 1 h and continued to do so throughout the remainder of the study. SCIv and Sham groups received daily equal amount of the PB mix without boldine.

### 2.4. Metabolomics and Lipidomic Sample Processing and Data Analysis

Blood was collected via ventricular puncture, allowed to clot at room temperature for 30 min, then spun down at 4 °C at 2000× *g* for 20 min. Serum was then collected and stored at −80 °C. 120 µL of serum was sent in two aliquots (60 µL per platform) on dry ice overnight to West Coast Metabolomics, a regional NIH Resource Core, at the University of California-Davis for untargeted primary metabolomics and complex lipidomics. All samples went through one freeze-thaw cycle that occurred during aliquoting for shipment. Metabolomics analyses were completed using reverse phase gas chromatography time-of-flight (TOF) mass spectroscopy in both positive and negative mode. Lipidomics was completed using quadrupole TOF in positive mode. Complete sample processing, data acquisition and data processing have been reported in detail [21]. Each individual platform was analyzed independently using Metabloanalyst 5.0 [22]. Data were first filtered to eliminate any feature with missing data (i.e., all samples had quantifiable data). Then both ‘omics platforms’ were filtered by selecting features with <40% relative standard deviation (RSD) of quality control samples. Data were visually checked for outliers using principal components analyses (PCA). Both ‘omics platforms’ were normalized to the respective feature median, log transformed, then pareto scaled. We have used these parameters to describe the metabolomics profile in post-SCI muscle tissue [4,5,23]. All spectra were matched to known metabolites using the BinBase algorithm [24] while unconfirmed molecules were matched to numerical BinBase IDs with identical spectra and retention times. Differences in individual features were tested using one-way ANOVAs with Benjamini-Hochberg false discovery rate (FDR) < 0.10 as the threshold for meaningful group differences. Tukey’s multiple comparisons post-testing was used when appropriate. Figures were generated using MetaboAnalyst 5.0 and the R package *complex_heatmap* (v2.13.1). The data matrices used to perform all lipidomic and metabolomic analyses can be found in Appendix A, respectively. Each respective table contains group means and standard deviations to better understand feature variability.

### 2.5. xMWAS

xMWAS [25] was used for multiomic integration using the framework of the *mixOmics* package [26] and multilevel community detection for network generation [27]. Features were selected using sparse projection of latent structures (sPLS) in regression mode. sPLS allows for simultaneous variable selection across both data matrices with better interpretability of constructed latent variables when compared to PLS [26]. The lipidomics data matrix was set as the ‘X matrix’ and was used to predict outcomes in the metabolomics data set (Y matrix). Features were filtered with an RSD < 0.40, with sPLS calculating pairwise correlations between data matrices. Statistical thresholds for selection were Pearson correlations > |0.40| and a *p* value < 0.05. The data sets used for xMWAS inputs were normalized and scaled as described above. Molecular communities were detected using the multilevel community detection algorithm. Absolute changes in eigenvector centrality (|ΔEIC|) were measured to find and compare important nodes within the network of each comparison, with |ΔEIC| > 0.30 set as a meaningful change in feature importance.

## 3. Results

### 3.1. Lipidomics

We identified 928 lipidomic features that were detected in every sample. RSD filtering removed 40 lipids, resulting in 888 lipids for further analyses. PCA, which maximizes variance to generate unsupervised loading scores, and sPLS-discriminant analysis (sPLS-DA), which maximizes supervised co-variance, led to unique clusters among groups (Figure 1A). Following one-way ANOVA analyses, 229 lipids had an FDR < 0.10, of which 92 lipids were annotated and 137 unannotated. The proportion of each annotated class is presented in Appendix A, with triglycerides (TGs, 30%), phosphatidylcholine (PC, 25%) and sphingomyelin (SM, 15%) being the most represented classes. No annotated long chain free fatty acid met FDR thresholds for differential abundance (Appendix A). The top 100 differentially abundant lipids based on FDR are shown as a heatmap in Figure 1B. Of the 229 differentially abundant lipids, 160 were similarly affected in magnitude and direction of change in both SCI groups compared to Sham (Appendix A). Of particular interest were lipids for which there were meaningful differences between SCIb and SCIv groups as determined using Tukey’s multiple comparisons post-test. There were 50 lipids that met these post-test criteria. Of these, 10 were TGs, 6 PCs, 4 phosphatidylethanolamines (PE), 4 phosphatidylinositols (PI), 1 ceramide (CER), 1 cholesterol ester (CE), and 24 that were not annotated. Figure 1C highlights the unique set of PIs that met FDR criteria that were altered by the administration of boldine.

### 3.2. Metabolomics

There were 953 metabolites detected in all samples. RSD filtering removed 111 metabolites, leaving 842 metabolites for analyses. PCA demonstrated overlap of the 95% confidence interval among all groups though the SCIv group was very tightly clustered. However, sPLS-DA was able to generate distinct group clusters (Figure 2A). 35 metabolites had a nominal *p* value < 0.05, though all had an FDR > 0.10. A heatmap of the top 50 metabolites as shown by nominal *p* value is shown in Figure 2B. For exploratory analyses of just the SCI groups, comparisons using independent samples *t*-tests resulted in 29 metabolites with a nominal *p* value < 0.05. While the SCIb vs. SCIv comparison found no metabolites with FDR < 0.10, consistent patterns for group differences were observed as shown by a heatmap of the 29 metabolites (Figure 2C). In a further exploratory analysis, we compared the list of 29 serum-based metabolites to the metabolites we previously found were altered by boldine in the skeletal muscle of these animals [5]. We found only one shared metabolite: an unannotated feature with BinBase ID 66261.

### 3.3. xMWAS

Multiomic integration revealed unique network structures and community membership for each group (Figure 3). Major outcomes of the network analyses are summarized in Table 1. sPLS selected 1112 features from the Sham group, with the network mapped across 8 communities with a modularity measure of 0.61. The SCIv group had 576 molecules selected and mapped across 7 communities with a modularity measure of 0.58 and the SCIb group had 887 features selected and mapped across 4 communities with a modularity measure of 0.33. Each community detected across all groups consisted mostly of unannotated lipids and metabolites (~60–70%) and a number of functional categories with low (1–5%) relative representation. However, in the Sham group, community 1 was 13% PCs, while communities 2 and 3 had 16% and 13% proportion of TGs, respectively. The SCIv group had two communities with >10% proportion of PCs: community 1 (17%) and community 3 (11%). The SCIb group had <10% for all annotated groups.

The major changes in network feature importance in the ‘SCIv vs. Sham’ comparison and ‘SCIv vs. SCIb’ comparisons are shown in Table 2. The ‘SCIv vs. Sham’ comparison resulted in 76 features with |ΔEIC| > 0.30 and the ‘SCIb vs. SCIv’ comparison resulted in 127 features with a |ΔEIC| > 0.30. The top features across both of these comparisons were mostly unannotated. Within the top annotated features with |ΔEIC| > 0.30 for the ‘SCIv vs. Sham’, multiple PCs were identified. However, within the ‘SCIv vs. SCIb’ comparison, the top annotated lipids were mostly SM, with additional changes in annotated metabolites related to sugars and amino acids.

## 4. Discussion

Conclusions supported by our data are that a mid-thoracic spinal cord transection results in large changes in the abundance of serum lipids with ~25% of all detected lipids affected by SCI 7 d after injury. Of the 28 TGs with an FDR < 0.10, all of them were downregulated in both SCI groups when compared to the Sham group, suggesting that either hepatic production of TGs is reduced or that peripheral breakdown of TG into free fatty acids is increased. Interestingly, boldine was able to attenuate the decrease in serum levels of 10 of these TGs (TGs 48:2, 50:1/2/4, 51:3, 52:4/5, 53:2/3/4). Of note, not a single free fatty acid had an FDR < 0.20, meaning despite major reductions in circulating TGs, free fatty acid levels were stable after SCI. Boldine was also able to lower serum levels of PI species to values similar to those observed in sham controls. The effects of SCI on the circulating metabolome were less clear with mean changes observed based on nominal *p* values, though none of these passed our FDR threshold. Taken together, SCI primarily affected the lipidome with some demonstrated efficacy of boldine able to reduce the magnitude or fully prevent SCI-induced changes in abundance.

Outside of the overall changes in the lipidome due to SCI, an interesting outcome of our study is the reduced circulating abundance of four PIs (PI 34:1, 34:2, 36:1 and 36:2) in the SCIb animals compared to SCIv, and to an extent, even the sham animals. These individual species have not been described after SCI or other neurological trauma but they have been studied in mice fed a high-fat diet. Serum PI 34:1 and 36:1 were upregulated in response to a chronic high-fat diet in mice and associated with elevated levels of blood glucose and pro-insulin [28], though the same animals had reduced levels of PI 34:2 and 36:2. While the literature implicates the alteration of PI levels in disrupting glucose metabolism through some PI mechanism, why boldine in particular would reduce these PI species after acute SCI is unknown. PIs are key components of the nuclear membrane, and present to a lesser extent in the plasma membrane [29]. One possibility is boldine was able to reduce accumulation of cytosolic calcium, leading to established intracellular mechanisms such as reduced PKC or phospholipase activity, which may affect turnover and packing of PI species [30]. Similarly, boldine was able to reduce abundance of a set of PCs compared to SCIv as well as sham animals (PC 34:0, 36:1, 37:6 and 38:1/2). PCs are the most abundant membrane phospholipids [31] suggesting that these phospholipids may be coming from either cell breakdown or increased membrane turnover. However, we are not able to identify where these phospholipids originated from our data.

Understanding the molecular relationships across multiomic profiles is key to identifying novel targets for therapeutics as well as repurposing existing interventions to improve clinical relevancy and translational potential. To improve our understanding of these relationships in our current report, we chose to use xMWAS for multiomic integration, community detection, and network analyses [25]. The network structure of the Sham group was relatively distinct compared to the SCI groups. Community detection across all groups consisted of mostly unannotated features and this is not surprising as our data sets were largely comprised of unannotated features. The SCIb group had a poor modularity index and large community membership, indicating that the algorithms had difficulty separating the community features. This is likely related to the highly variable changes seen in the metabolome in these animals. We next compared EIC among groups to determine whether the importance of key features was changed in our main comparisons of ‘SCIv vs. Sham’ and ‘SCIb vs. SCIv’. EIC represents the weighted importance of a feature (i.e., number of connections) and its connectedness to other important features. The majority of the features with large changes in EIC were, again, mostly unannotated. However, there were a number of annotated features that met our statistical cutoff of a |ΔEIC| > 0.30, which, while arbitrary, is more conservative than the cutoff (|ΔEIC| > 0.10) for the original xMWAS publication [25]. When comparing the differences in EIC between annotated features selected in each comparison, the top lipids in the ‘SCIv vs. Sham’ comparison were mostly PCs, while the ‘SCIb vs. SCIv’ comparison was mostly SM. The changes in EIC related to SM species in the SCI animals are interesting as SM are key components of the membrane of myelinating cells (e.g., oligodendorcytes). SM species are intricately linked to calcium signaling both as being upstream initiators of signaling and downstream mediators [32]. Another intriguing outcome in the ‘SCIv vs. SCIb’ comparison was that the top annotated metabolites were sugars and amino acids. Proline, glucose, and phenylalanine were molecules we noted to be differentially abundant following boldine treatment in paralyzed skeletal muscle from these animals [5], as was glutamine, a well-known regulator of neuronal function that is dysregulated after SCI [33]. The lack of molecular annotations for important features identified using xMWAS results in our inability to fully appreciate the change in molecular landscape. However, it does highlight key biological process affected by SCI with or without boldine, providing direction for future studies to improve translational potential.

A major limitation of untargeted approaches for lipidomic and metabolomic studies is the lack of annotation for a large portion of molecules that get detected. In this report unannotated features make up ~60% of detected lipids and ~75% of detected metabolites. Among the differentially abundant lipids that met FDR criteria, ~60% were unannotated, making complete understanding of our data difficult. While this may complicate complete biological understanding, it is clear the magnitude of change observed in the lipidome highlights a unique response to, and metabolic challenge during, acute SCI in mice. Another limitation of our study was the inability to detect any group differences in metabolomics at our FDR threshold. This was undoubtedly due to small sample size and the fact that two animals in both the SCIb and Sham groups drove a large majority of the variance as we noted in Figure 2A. While it could be justified these mice could have been outliers to be removed from analyses, we could not identify any apparent technical or sample processing issues that may have led to our variable results and, notably, identical serum aliquots were used for the lipidomic analyses, which showed no major outliers. Due to our already small sample sizes, we felt it was inappropriate to remove any of the animals despite them likely being outliers. Additional limitations to our approach were potential unknown interactions of DMSO and/or the high oil content of our vehicle in SCI animals compared to the sham controls, as well as not recording exact amounts of food and water intake or having animals explicitly fasted for a period of time before euthanasia.

In closing, our data clearly demonstrated the circulating lipidome is greatly affected by acute SCI, and boldine was associated with preserving the levels of a subset of TGs, PEs, PIs and PCs. Due to the nature of untargeted mass spectrometry, a large portion of our data are unannotated, limiting complete and in-depth biological understanding of how SCI affects the lipidome and metabolome. Additionally, while the metabolomic variability of the SCIb and Sham animals resulted in minimal statistical differences among groups, some differences at the nominal level were observed. We anticipate future studies will clarify these outcomes.

## Figures and Tables

**Figure 1 ijerph-20-06591-f001:**
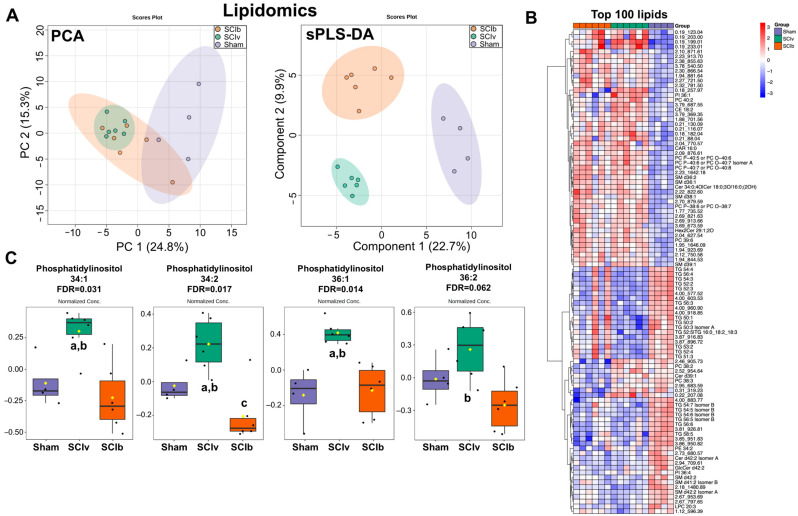
Changes in the circulating lipidome after SCI. (**A**) PCA and sPLS-DA plots showing unique clustering of groups across the top two components for each respective plot. (**B**) Clustering heatmap of the top 100 differentially abundant lipids (FDR < 0.10) showing mainly SCI-induced changes compared to sham controls. (**C**) Unique and consistent boldine-associated changes in differentially abundant circulating phosphatidylinositols. PCA and sPLS-DA plots are shown with 95% confidence interval range. Box plots are median values that have been log transformed and pareto scaled, with the yellow diamond equaling the mean. Statistically different Tukey’s multiple comparison post-tests are denoted with ‘a’ = *p* < 0.05 compared to ‘Sham’, ‘b’ = *p* < 0.05 compared to ‘SCIb’ and ‘c’ = *p* < 0.05 compared to ‘Sham’.

**Figure 2 ijerph-20-06591-f002:**
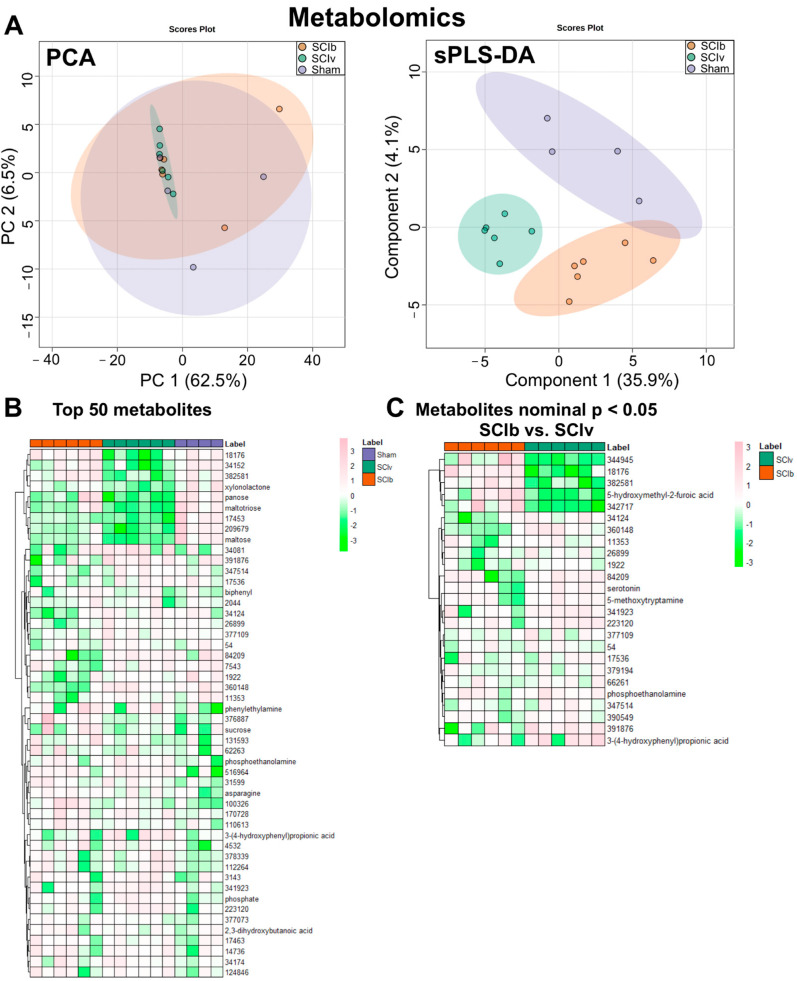
Metabolomics of serum after 7 d after a complete spinal injury. (**A**) PCA plot showing major overlap of the ‘Sham’ and ‘SCIb’ groups, with the sPLS-DA plot showing unique clusters. (**B**) Clustering heatmap of the top 50 metabolites as ranked by nominal *p* value from one-way ANOVAs. (**C**) Clustering heatmap of metabolites with a nominal *p* value < 0.05 in exploratory comparisons of ‘SCIv’ and ‘SCIb’ mice.

**Figure 3 ijerph-20-06591-f003:**
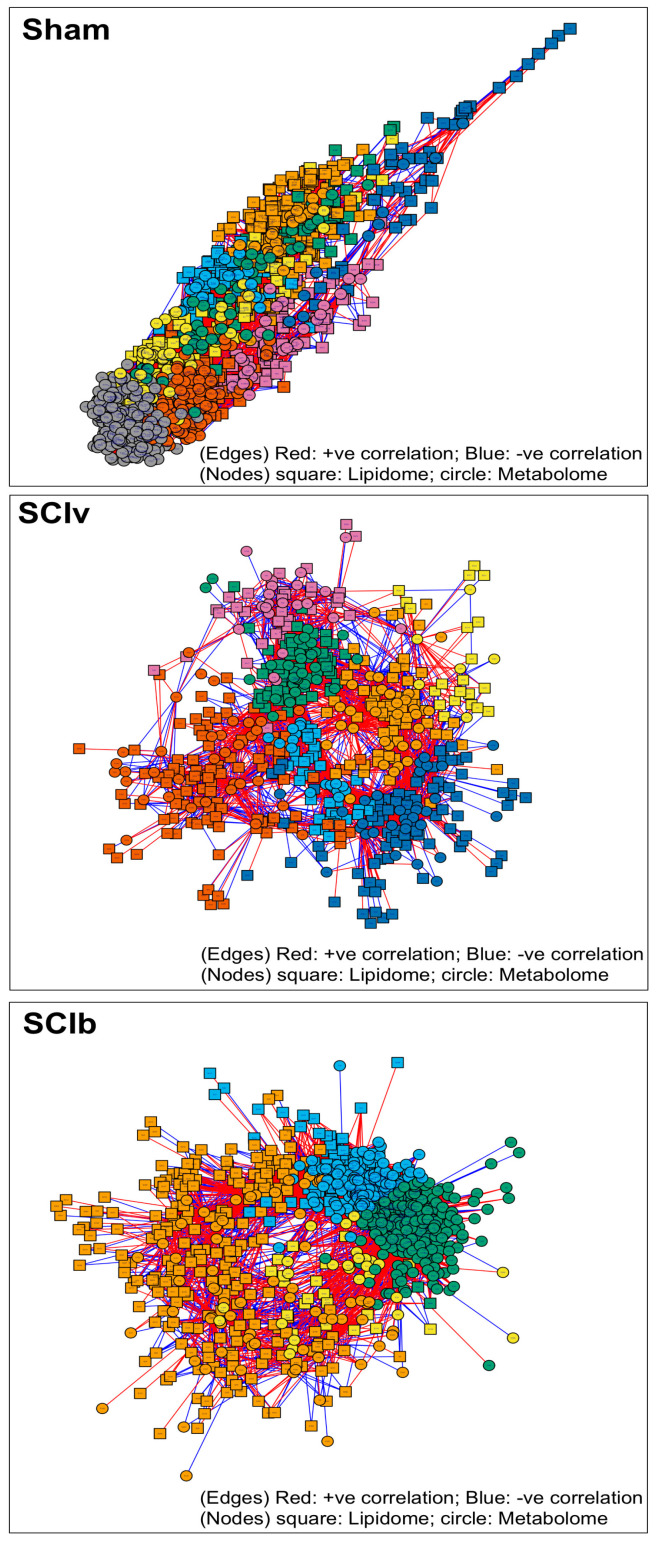
Multiomic integration and network generation using xMWAS. Features were selected using sPLS in regression mode with correlations > |0.40| and *p* values < 0.05. Distinct communities as generated by the multilevel community detection algorithm are shown in different colors for each respective network without a shared color arrangement across networks (i.e., an orange community in one network is not equivalent to an orange community in another). Red edges are positive correlations between nodes, with blue edges being negative. Lipids are shown as square symbols and metabolites as circles.

**Table 1 ijerph-20-06591-t001:** Community features and membership proportions following xMWAS.

xMWAS Data Summary				
Sham				
	*Total features*	*Number of annotated groups*	*Top community proportion (%)*	*Top Annotated Proportion (%)*
Community 1	189	13	Unannotated lipids (56%)	Phosphatidylcholines (13%)
Community 2	75	10	Unannotated lipids (45%)	Triglycerides (16%)
Community 3	116	16	Unannotated metabolites (31%)	Triglycerides (13%)
Community 4	143	17	Unannotated metabolites (38%)	Triglycerides (6%)
				Phosphatidylcholines (6%)
Community 5	56	9	Unannotated lipids (50%)	Phosphatidylcholines (21%)
Community 6	166	15	Unannotated metabolites (59%)	Sugars (4%)
Community 7	104	10	Unannotated lipids (54%)	Phosphatidylcholines (7%)
Community 8	262	14	Unannotated metabolites (83%)	Amino acids and biogenic amines (4%)
Matrix modularity	0.61			
SCIv				
Community 1	94	12	Unannotated lipids (41%)	Phosphatidylcholines (17%)
Community 2	56	8	Unannotated lipids (59%)	Phosphatidylcholines (4%)
Community 3	89	12	Unannotated lipids (40%)	Phosphatidylcholines (11%)
Community 4	27	6	Unannotated lipids (67%)	Phosphatidylcholines (4%)
				Phosphatidylethanolamines (4%)
				Sphingomyelin (4%)
				Amino acids and biogenic amines (4%)
Community 5	119	11	Unannotated lipids (48%)	Phosphatidylcholines (7%)
Community 6	136	18	Unannotated lipids (51%)	Phosphatidylcholines (6%)
Community 7	55	10	Unannotated lipids (42%)	Amino acids and biogenic amines (7%)
				Phosphatidylcholines (7%)
Matrix modularity	0.58			
SCIb				
Community 1	294	20	Unannotated lipids (45%)	Triglycerides (5%)
Community 2	244	19	Unannotated metabolites (48%)	Phosphatidylcholines (7%)
Community 3	302	15	Unannotated metabolites (77%)	Amino acids and biogenic amines (4%)
Community 4	47	11	Unannotated metabolites (45%)	Triglycerides (6%)
Matrix modularity	0.33			

**Table 2 ijerph-20-06591-t002:** Changes in eigenvector centrality for each of the main group comparisons.

xMWAS Features with |ΔEIC| > 0.30							
SCIv vs. Sham							
*Top lipids*	*|ΔEIC|*	*Top annotated lipids*	*|ΔEIC|*	*Top metabolites*	*|ΔEIC|*	*Top annotated metabolites*	*|ΔEIC|*
PC 37:2	1.00	PC 37:2	1.00	BinBase ID 64546	1.00	sedoheptulose 7-phosphate	0.38
1.96, 640.59 (rt, m/z)	0.97	SM 46:6	0.87	BinBase ID 390122	0.99	spermidine	0.38
2.90, 922.01 (rt, m/z)	0.97	CER d34:0	0.84	BinBase ID 379194	0.95	urea	0.34
0.93, 482.40 (rt, m/z)	0.94	PE P-36:1 or PE O-36:2	0.66	BinBase ID 42357	0.87		
0.19, 135.01 (rt, m/z)	0.9	LPC 16:0	0.66	BinBaseID 3173	0.83		
SM 46:6;2O	0.89	PC 40:7	0.65	BinBase ID 210272	0.80		
CER d34:0	0.84	PC 39:6	0.64	BinBase ID 7542	0.72		
0.19, 153.02 (rt, m/z)	0.78	PC 40:8	0.62	BinBase ID 16792	0.71		
0.62, 522.35 (rt, m/z)	0.67	PC 40:5 Isomer B	0.6	BinBase ID 4794	0.65		
PE P-36:1 or PE O-36:2	0.66	PC 40:5 Isomer A	0.59	BinBase ID 120987	0.65		
SCIb vs. Sham							
*Top lipids*	*|ΔEIC|*	*Top annotated lipids*	*|ΔEIC|*	*Top metabolites*	*|ΔEIC|*	*Top annotated metabolites*	*|ΔEIC|*
0.18, 257.97 (rt, m/z)	1.00	SM d42:2 Isomer A	0.88	BinBase ID 42357	0.84	mannose	0.44
2.07, 952.59 (rt, m/z)	0.97	SM d38:1	0.86	BinBase ID 3173	0.79	creatinine	0.44
2.36, 882.62 (rt, m/z)	0.95	LPC 16:0	0.65	BinBase ID 379194	0.77	valine	0.43
2.16, 936.60 (rt, m/z)	0.93	SM d41:2 Isomer B	0.64	BinBase ID 4794	0.65	glucose	0.42
1.72, 689.54 (rt, m/z)	0.89	PC 40:8	0.62	BinBaseID 120987	0.64	phosphate	0.42
1.46, 764.55 (rt, m/z)	0.89	SM d40:2 Isomer B	0.60	BinBase ID 64546	0.80	proline	0.41
SM d42:2 Isomer A	0.88	PC 40:5 Isomer A	0.59	BinBase ID 7542	0.56	ribose	0.41
1.95, 908.57 (rt, m/z)	0.86	PC 40:4	0.56	BinBase ID 390122	0.55	phenylalanine	0.40
1.33, 762.53 (rt, m/z)	0.86	SM d41:1	0.50	BinBase ID 342919	0.51	sedoheptulose 7-phosphate	0.38
SM d38:1	0.86	SM d42:1	0.48	BinBase ID 161365	0.46	glutamine	0.36

Abbreviations: CER = ceramide, EIC = eigenvector centrality, LPC = lyso-phosphatidylcholine, PC = phosphatidylcholine, PE = phosphatidylethanolamine, SM = sphingomyelin, rt = retention time, m/z = mass:charge.

## Data Availability

Raw and processed metabolomics and lipidomics data matrices are presented in the Appendix A. Statistical outcomes for all analyses are available in the Appendix A. Data related to mice are available from our previous publication [5].

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
