# Peer review of "Boldine Alters Serum Lipidomic Signatures after Acute Spinal Cord Transection in Male Mice"

_ijerph, 2023, doi:10.3390/ijerph20166591_

Round 1

Reviewer 1 Report

Manuscript ID: ijerph-2468738

Manuscript Title: Boldine alters serum lipidomic signatures after acute spinal cord transection in male mice

Authors investigated the effects of boldine on serum lipid and metabolic profiles 7 days after spinal cord transection. The results are interesting, but there are some concerns to consider for publication.

Several studies demonstrated the effects of boldine on axonal sprouting and gastrocnemius muscle 28 or 56 days after spinal cord transection. However, authors conducted 7 days after a complete spinal cord transection. Authors should clearly demonstrate why they used the serum samples from 7 days after a complete spinal cord transection, and 4-month-old C57BL/6 mice in this study. In addition, authors should show the composition of chow diet used in this study.

Authors used DMSO and peanut oil as a vehicle. Are there any effects on the regeneration process in spinal cord or lipid profiles in serum?

Authors described that they published the data about weight and muscle loss. Because authors analyzed the lipidomics and metabolomics, authors should show the food and water intake in the study. In addition, authors used 4-6 animals per group and I wonder the differences on food intake in sacrificing period because authors supplied the diet with ad libitum.

Author Response

We thank the reviewers for their time and effort to reviewer our manuscript and provide feedback. We are encouraged by the overall positive reviews of the manuscript. We have addressed each comment below, with our response in bold underneath. Major changes to the manuscript are noted with yellow highlight. Minor changes (typos, extra characters, etc.) were corrected without note. We hope the revisions made in response to your suggestions and comments have resulted in a stronger manuscript and is now suitable for publication.

Reviewer 1:

Authors investigated the effects of boldine on serum lipid and metabolic profiles 7 days after spinal cord transection. The results are interesting, but there are some concerns to consider for publication.

  1. Several studies demonstrated the effects of boldine on axonal sprouting and gastrocnemius muscle 28 or 56 days after spinal cord transection. However, authors conducted 7 days after a complete spinal cord transection. Authors should clearly demonstrate why they used the serum samples from 7 days after a complete spinal cord transection, and 4-month-old C57BL/6 mice in this study. In addition, authors should show the composition of chow diet used in this study.

Thank you for this comment. We selected 7 days post spinal cord transection for our analysis because this is the timepoint at which we observe large-scale changes in the metabolome and transcriptome of paralyzed gastrocnemius muscle (Potter et al. 2023). In fact, the metabolomic signature was not at all altered 28 d post-SCI, which we have demonstrated in two separate papers (Graham et al. 2019, Potter et al 2023). Thus, we reasoned that there may be robust changes in serum profiles at a time when large changes in skeletal muscle profiles were also seen. We chose C57BL6 mice because they tolerate SCI well, are a common background for genetically modified mice, and all of our prior studies with boldine have used C57BL6 mice. We use mice at between 3 to 5-months-old as younger mice have poorer survival after SCI. Composition of micro-nutrient control chow used in the study (Research Diets, D12450J) is as follows (10% fat, 20% protein, 70% carbohydrate, 3.82 kcal/g energy density). We have made these suggested edits in lines 101-103, 103-105, and 122-123.

  1. Authors used DMSO and peanut oil as a vehicle. Are there any effects on the regeneration process in spinal cord or lipid profiles in serum?

And

  1. Authors described that they published the data about weight and muscle loss. Because authors analyzed the lipidomics and metabolomics, authors should show the food and water intake in the study. In addition, authors used 4-6 animals per group and I wonder the differences on food intake in sacrificing period because authors supplied the diet with ad libitum.

This is a great question. DMSO ingestion with or without peanut oil/peanut butter could potentially alter circulating metabolomic and lipidomic signatures. However, the experimental design eliminated this potential confound because all animals received identical diets, including peanut butter. While DMSO has been linked with neuroprotective properties, our complete transection model severs the spinal cord such that regeneration is unable to restore connections between supraspinal centers (sensorimotor cortex, brainstem and others) and greatly reduces the likelihood of this occurring. And as all animals received it mixed with peanut butter/oil, our experimental design further increases the rigor of the study. We did not evaluate any markers for neuroplasticity in spinal cord above or below the site of the transection though animals remained completely paralyzed for the duration of the original 28 d study. Any observable changes in neuroplasticity followed our complete transection model would likely not be observed 7 d post-surgery.

We have previously described in detail food intake, in our hands, using C57Bl6 SCI mice (Liu, et al, 2021). These animals eat around 4-5 g of diet chow per day and drink between 4 and 6 ml of water. We did not measure food consumption or water intake in the current study. Mice eat all peanut butter offered to them from about day 5 after SCI on suggesting that it is unlikely that differences in lipids reflect differences in dietary intake. All groups received an equal amount of peanut butter. Peanut butter contains primarily oleic acid, and palmitic acid (Parilli-Moser et al, 2022). Levels of these lipids were not different between groups adding to the evidence that peanut butter was not a confound.   

We have added these discussed these considerations in the revised manuscript (lines 335-338).

Reviewer 2 Report

Metabolomics is a field, categorised as systems biology, that encompasses an interdisciplinary field of knowledge focusing on the explanation of changes occurring in the metabolic profile. This profile consists of a variety of low-molecular compounds such as lipids, organic acids, carbohydrates, aminoacids, nucleotides or steroids. Metabolic profile analysis has most often been performed in cancer, nephrotoxicity, neurodegenerative  and psychiatric diseases. The use of metabolomic and lipidomic analysis of the serum of mice treated with boldin or vehicle is an original research and possibly therapeutic method with the potential to improve the condition of patients after spinal cord injury.

The reviewer has minor comments on the content of the article. The order of the information provided is in inappropriate  places. Among other things, the results are given in the introduction and the conclusions at the beginning of the discussion. The introduction lacks a description of the theoretical basis of metabolomics and its application in diagnosis and therapy.

The authors do not specify in which form the Ringer's fluid, analgesics and the mixture were administered. It is also not stated whether the animals received any other food. It is also not described how the animals left alone for 7 days behaved and what happened to them after the test.

The statistical results obtained are valuable and may suggest a direction for further studies, including clinical research. The reviewer suggests that there is no description of the clinical condition of the tested animals in comparison with the obtained metabolomic data, which might be useful in the analysis of laboratory results. The researchers do not propose the applicability of the obtained data in practice.

Author Response

Reviewer responses for manuscript IJERPH 2468738 R0

We thank the reviewers for their time and effort to reviewer our manuscript and provide feedback. We are encouraged by the overall positive reviews of the manuscript. We have addressed each comment below, with our response in bold underneath. Major changes to the manuscript are noted with yellow highlight. Minor changes (typos, extra characters, etc.) were corrected without note. We hope the revisions made in response to your suggestions and comments have resulted in a stronger manuscript and is now suitable for publication.

Reviewer 2

Metabolomics is a field, categorised as systems biology, that encompasses an interdisciplinary field of knowledge focusing on the explanation of changes occurring in the metabolic profile. This profile consists of a variety of low-molecular compounds such as lipids, organic acids, carbohydrates, aminoacids, nucleotides or steroids. Metabolic profile analysis has most often been performed in cancer, nephrotoxicity, neurodegenerative  and psychiatric diseases. The use of metabolomic and lipidomic analysis of the serum of mice treated with boldin or vehicle is an original research and possibly therapeutic method with the potential to improve the condition of patients after spinal cord injury.

The reviewer has minor comments on the content of the article. The order of the information provided is in inappropriate  places. Among other things, the results are given in the introduction and the conclusions at the beginning of the discussion. The introduction lacks a description of the theoretical basis of metabolomics and its application in diagnosis and therapy.

We thank the reviewer for the comment. The organizational critique is simply a difference in writing style and preference. We feel it is appropriate to have a small outline of the main outcomes of the study at the end of the introduction and a brief summary as the first paragraph of the discussion. As there are no specific formatting restrictions for the journal we feel our structure is consistent with journal requirements.

In regards to the theoretical basis of using metabolomics, we have added a small section to the introduction explaining this rationale (lines 40-45).

The authors do not specify in which form the Ringer's fluid, analgesics and the mixture were administered. It is also not stated whether the animals received any other food. It is also not described how the animals left alone for 7 days behaved and what happened to them after the test.

Thank you for this comment. We have previously described in detail our post-surgical methods and the characteristics of the animals over the first 7 days after spinal cord transection and have them appropriately cited within the manuscript (Graham, 2020; Liu, 2021; Toro,  2021; Toro, 2023; Potter, 2023). Additionally, we have added we delivered the analgesics via subcutaneous injection (lines 135 and 136). As the handling of these animals was described in detail in our previous paper as we cite in the manuscript (Potter, 2023), we feel this section is described is sufficient.

The statistical results obtained are valuable and may suggest a direction for further studies, including clinical research. The reviewer suggests that there is no description of the clinical condition of the tested animals in comparison with the obtained metabolomic data, which might be useful in the analysis of laboratory results. The researchers do not propose the applicability of the obtained data in practice.

Effects of the spinal cord transection on body weight and muscle weights were reported in our recent paper on effects of boldine on transcriptomics and metabolomics outcomes (Potter, 2023). Spinal cord transected animals demonstrate complete paralysis of hindlimbs consistent with our prior reports (Graham 2020, Liu 2021, Potter 2023). We agree that it will be important to consider effects of boldine on whole body metabolism in future pre-clinical studies and, if warranted, in clinical studies as well. We have expanded our discussion to include the points in lines 300-303 and 329-331
